# Scale up and strengthening of comprehensive emergency obstetric and newborn care in Tanzania

**Angelo S. Nyamtema**[1,2]*, **John C. LeBlanc**[3], **Godfrey Mtey**[1], **Gail Tomblin Murphy**[4], **Elias Kweyamba**[1,2], **Janet Bulemela**[1,2], **Allan Shayo**[1], **Zabron Abel**[1], **Omary Kilume**[1,2], **Heather Scott**[3], **Janet Rigby**[4]

**1** Tanzanian Training Centre for International Health, Ifakara, Tanzania, **2** St. Francis University College for Health and Allied Sciences, Ifakara, Tanzania, **3** Dalhousie University, Halifax, Nova Scotia, Canada, **4** WHO/PAHO Collaborating Centre on Health Workforce Planning & Research, Dalhousie University and Nova Scotia Health, Halifax, Nova Scotia, Canada

* nyamtema_angelo@yahoo.co.uk

**Data Availability Statement:** All relevant data are within the paper and Supporting Information files.

**Funding:** This work was carried out with the aid of grants number 108027 & 108548 from the

## Abstract

### Introduction

In Tanzania, inadequate access to comprehensive emergency obstetric and newborn care (CEmONC) services is the major bottleneck for perinatal care and results in high maternal and perinatal mortality. From 2015 to 2019, the Accessing Safe Deliveries in Tanzania project was implemented to study how to improve access to CEmONC services in underserved rural areas.

### Methods

A five-year longitudinal cohort study was implemented in seven health centres (HCs) and 21 satellite dispensaries in Morogoro region. Five of the health centres received CEmONC interventions and two served as controls. Forty-two associate clinicians from the intervention HCs were trained in teams for three months in CEmONC and anaesthesia. Managers of 20 intervention facilities, members of the district and regional health management teams were trained in leadership and management. Regular supportive supervision was conducted.

### Results

Interventions resulted in improved responsibility and accountability among managers. In intervention HCs, the mean monthly deliveries increased from 183 (95% CI 174–191) at baseline (July 2014 –June 2016) to 358 (95% CI 328–390) during the intervention period (July 2016 –June 2019). The referral rate to district hospitals in intervention HCs decreased from 6.0% (262/4,392) with 95% CI 5.3–6.7 at baseline to 4.0% (516/12,918) with 95% CI 3.7–4.3 during the intervention period while it increased in the control group from 0.8% (48/5,709) to 1.5% (168/11,233). The obstetric case fatality rate decreased slightly from 1.5% (95% CI 0.6–3.1) at baseline to 1.1% (95% CI 0.7–1.6) during the intervention period (not

Innovating for Maternal and Child Health in Africa initiative- a partnership of Global Affairs Canada (GAC), the Canadian Institutes of Health Research (CIHR) https://cihr-irsc.gc.ca/e/193.html and Canada's International Development Research Centre (IDRC) https://www.idrc.ca/en. The funders had no role in study design, data collection and analysis, decision to publish, or preparation of the manuscript.

**Competing interests:** The authors have declared that no competing interests exist.

statistically significant). Active engagement strategies and training in leadership and management resulted in uptake and improvement of CEmONC and anaesthesia curricula, and contributed to scale up of CEmONC at health centre level in the country.

## Conclusions

Integration of leadership and managerial capacity building, with CEmONC-specific interventions was associated with health systems strengthening and improved quality of services.

## Introduction

The seeds of improving maternal and newborn health were planted in Tanzania following the 1987 birth of the Safe Motherhood Initiative (SMI) in Nairobi accompanied by an international call to action [1]. Following that, the government of Tanzania established the Safe Motherhood Unit within the Ministry of Health. Several interventions were then identified and included in the national policies and plans, including: strengthening the health system to provide skilled attendance during childbirth, upgrading rural health centres to provide emergency obstetric and newborn care (EmONC) services, providing adolescent and male friendly family planning services, strengthening public–private partnership to ensure continuum of care, and strengthening community participation [2, 3]. Despite a number of interventions, maternal mortality ratio stagnated above 500 per 100,000 livebirths (specifically in 1999, 2005 and 2015 it was estimated to be 528, 578 and 556 per 105 live births, respectively) [4–6]. The major bottlenecks identified included -inadequate and ineffective implementation approaches of these services because of gaps in leadership and management (L&M) at all levels of the health system, coupled with inadequate material, financial and human resources [2, 7, 8].

Success stories in reducing maternal and newborn deaths exist in Tanzania and across the globe due to integrating leadership and management, and medical-based interventions for maternal and newborn care [9–13]. Evidence indicates that when maternal and child health interventions lead to improved quality of obstetric and newborn care, the success has been attributed to strong leadership in reproductive health and accountability of health providers, managers and key decision makers as well as the presence of enabling policies [9]. In Kigoma, Tanzania, transformational leaders and strong managers of locally available material and human resources strengthened maternal services at a regional hospital. This resulted in reduction of maternal mortality ratio from 933 to 186 per 105 live births over the period 1984–91 [11, 12].

In 2015, as part of the Innovating for Maternal and Child Health in Africa (IMCHA) initiative funded by Canada's International Development and Research Centre, the Accessing Safe Deliveries in Tanzania (ASDIT) project was designed to study how to improve access to comprehensive emergency obstetric and newborn care (CEmONC) services in underserved rural Tanzania, where only 12% of HCs were then providing CEmONC [3]. CEmONC services are the key medical interventions that are used to treat direct obstetric complications that cause the vast majority of maternal deaths around the globe [14]. Implementation of the project revealed weaknesses in leadership and management as well as high rates of maternal and perinatal adverse outcomes. Most health managers at the primary health facilities, district and regional levels had inadequate leadership and management skills. These findings formed the basis for integrating the ASDIT project medical-based interventions with a sister program for

leadership and managerial capacity strengthening for quality pregnancy and newborn care. This paper presents the overall key findings from these programs.

## Materials and methods

### Study settings, design and public involvement

At the start of the study, Morogoro region had 15 HCs that were either already offering CEmONC or would be able to do so once staff are trained. This study was a longitudinal cohort study in seven health centres in Morogoro region, Tanzania. Five of these received an intervention and two served as controls to detect secular trends. The HCs in the intervention group were Kibati, Ngerengere, Gairo, Melela and St. Joseph HCs. The hierarchy of health facilities in Tanzania, from bottom to top, includes dispensaries, health centres (HCs), district hospitals, regional hospitals, zonal hospitals and the national specialized hospitals. By design, study health centres had to be far enough from the nearest referral hospital for referral to be a significant challenge for health centre staff and families. Health centres were also chosen to reflect the diversity of funding and governance models for HCs in Tanzania. As described elsewhere, Kibati and Ngerengere HCs had the proper infrastructure for CEmONC services including maternity and neonatal wards, operating theatre and ability to provide blood transfusion but their staff had not received CEmONC training [15]. This group typified the HCs that the government would have to upgrade as it implemented its national goal of 50% of HCs in Tanzania offering CEmONC services by 2020 [3]. Gairo and Melela HCs (publicly funded), and St. Joseph HC (representing a group of faith-based organizations) were already providing CEmONC and were included in the intervention group to study how CEmONC services could be strengthened. There were therefore 5 health centres that underwent the intervention. Mlimba and Mkamba HCs were randomly allocated to the control group from the remaining 5 publicly funded HCs that were already providing CEmONC services. Because of the intentional differences of intervention and control HCs, this study was designed as a before-after intervention where different funding and governance models could be compared. Comparisons of intervention and control HCs are mainly to detect secular trends that could potentially explain the before-after differences observed in the intervention HCs.

The study was conceived and designed by the ASDIT team, a multidisciplinary group of researchers at the Tanzanian Training Centre for International Health (Tanzania) and Dalhousie University (Canada), partnering with Morogoro regional administration representing the varied interests of patients, health care providers, healthcare systems and policy makers. The Regional Medical Officer for Morogoro region (GM) was engaged as a public co-investigator and worked as a liaison between the district, regional and national authorities. To identify the facilities, most relevant research topics and meaningful outcomes, we worked with the public co-investigator and administered a leadership and management (L&M) survey customized to care providers. Through workshops and meetings, the project team regularly shared findings with the key stakeholders at district, regional and national levels to provide them with a broader understanding of the project, and the progress and outcomes.

### The theory of change: A model formulation

In order to develop a set of sound and scientifically derived interventions the project applied principles of operations research to identify and address operational factors that determine maternal and newborn health care in Tanzania [16, 17]. Using evidence-based science on the interventions that work, available material, financial and human resources the project blended medical-based, and leadership and managerial interventions (Fig 1) [18, 19].

| PROBLEM | FACTORS FOR CHANGE | INPUTS | OUTPUTS | IMPACT |
|---|---|---|---|---|
| High maternal and newborn deaths & morbidity | Gaps in health systems structure and resources | Strengthening L&M in MCH<br><br>Strengthening skills in EmONC<br><br>Strengthening anaesthesia services<br>Introduce CEmONC at health centres | Improved access to CEmONC services<br><br>Reduced referrals to secondary facilities | Improved maternal and newborn health (reduced mortality and complications) |

*Note: L&M = leadership and management; MCH= maternal and child health*

**Fig 1. The theory of change for improved maternal and newborn health care in the ASDIT project.**

### Interventions

**Capacity building in emergency obstetric and newborn care and anaesthesia.** Forty-two associate clinicians from the five intervention HCs were trained in teams for three months in CEmONC and anesthesia. Considering the national regulations, assistant medical officers (advanced associate clinicians) from these HCs were trained in CEmONC while clinical officers and nurse-midwives (associate clinicians) were trained in anaesthesia, postoperative care and care of the sick and premature newborn [15]. In Tanzania, assistant medical officers are licensed to perform surgery. The two years of training includes three months in general surgery and three months in obstetrics and gynaecology. The lack of internship program and inadequate supervision after graduation denies them the opportunity to acquire adequate surgical skills in obstetrics. This CEmONC training program was designed to strengthen surgical skills taking into consideration that they were expected to work independently in remote HCs. The curricula for CEmONC and anaesthesia were built on training programs for associate clinicians previously delivered at the St. Francis Referral Hospital [20].

To reinforce knowledge and skills, post-training activities included eHealth strategies, quarterly supportive supervision visits and continuous mentorship. The eHealth strategies included the offline eLearning modules and tele-consultation. For tele-consultation, care providers at the intervention HCs were also linked with obstetricians, a paediatrician and an anaesthetist based at St. Francis Referral Hospital. Mentorship and supportive supervisory visits were done every three months and included clinical audits and data collection for C-sections, maternal deaths and morbidities, fresh stillbirths, early neonatal deaths and methods of anaesthesia [15]. Mentorship activities focused on identified areas of substandard care.

**Strengthening leadership and management.** The project team designed capacity-building workshops in L&M and onsite mentorship geared at equipping health managers with essential knowledge and skills on leading change. These were basic principles and strategies in leading change that would improve performance and CEmONC services at their health facilities. The workshops were conducted in 2018 and 2021 and involved participants from 20 primary health facilities, i.e., the 5 intervention health centres and 15 satellite dispensaries, members of the district and regional health management teams specifically the district medical officers and other district health personnel. These dispensaries referred patients with medical

complications to their respective health centres. Quality improvement plans developed after the 2018 "Big Results Now" (BRN) Star Rating assessment were used to mentor (onsite) the health facility health management teams and jointly address the gaps identified. The BRN star rating is a government system that measures the performance of various healthcare facilities aimed at improving quality of healthcare [21]. Since prior research has shown that engaging workplace teams in leadership development programmes is critical to success [22], onsite mentorship was a major component of the ASDIT intervention.

## Data collection

All data were collected by the research team. The data on CEmONC services were collected concurrently with supportive supervisory visits as described above. The data included deliveries, types of anaesthesia, referrals and audit results of pregnancy adverse outcomes (maternal and perinatal morbidity and mortality). These were obtained from the working log books at each centre. Data on L&M were collected at baseline in 2018 and at the end of the study in 2021 using validated tools i.e., the "Big Results Now" Star Rating assessment and L&M survey tools [21, 23]. The BRN tool assesses the following domains: 1) health facility management (12 indicators); 2) use of facility data for service improvements (6 indicators); 3) staff performance management (5 indicators); 4) organization of services (8 indicators); 5) handling of emergencies/referral (7 indicators); 6) client focus (4 indicators); 7) social accountability (7 indicators); 8) facility infrastructure (14 indicators); 9) infection prevention and control (11 indicators); 10) clinical services (13 indicators); and 11) clinical support services (23 indicators) (S1 Table).

The BRN system rates health facilities from 0 to 5 stars depending on the quality of services provided. The BRN star rating is based on the score of the minimum scoring domain and not the total or average marks. A score of 0–19% is graded no stars, 20–39% one star, 40–59% two stars, 60–79% three stars, 80–89% four stars, and 90–100% five stars [21]. The target of the government improvement initiative was to have 80% of primary health facilities rated with three stars or more by 2017–18. Three stars implied that the facility was performing at a minimum required standard and the domain scoring the least scored 60–79%.

The "Leadership and Management" (L&M) survey primarily used Likert scales to assess data on care providers' perceptions on L&M competencies, focusing on the following domains: team climate of facilities; staff role clarity; and job satisfaction (Table 1). CEmONC costs were collected from the health centres, Tanzania Medical Store Department and non-governmental organizations that had upgraded health centres for CEmONC services provision.

**Table 1. Leadership and managerial domains assessed using L&M survey in 2018 and 2021.**

1. Domain: Team climate of facilities
   Involvement of staff in the process of setting the objectives, clarity, achievement and worthwhile of the objectives to the health care facility, and information sharing, Team climate is defined as a healthy, supportive and engaging environment for employees at the workplace
2. Domain: Staff role clarity
   Clarity of individual roles/ responsibilities, planned goals and objectives, existence of performance targets, expectations, adequacy of resources to support implementation of the assignments and ability to meet annual performance targets.
3. Domain: Job satisfaction
   Satisfaction with communication across the facility, control over given job activities, expectations of the job done, involvement in decision-making processes, facility's support for individual learning and development, safety of work environment, the balance between work and family/personal life, opportunities for social contact at work, opportunities to interact with management/administration, amount of responsibility.

## Uptake strategies

Several uptake strategies were employed to enhance uptake of key interventions. These included engagement of key decision makers and the regional and council management teams throughout the project implementation period. The team conducted biannual national and regional stakeholders' meetings and provided updates of the project during the quarterly regional maternal and child mortality audit meetings.

## Data analysis

Using Stata (version 15), one-way ANOVA and Chi-square tests were used to assess the impact of the intervention model by comparing outcomes during the baseline (July 2014—June 2016) and intervention (July 2016—June 2019). A one-way ANOVA test was used to determine the statistical differences of the mean monthly deliveries and mean score of the BRN key domains. Chi-square tests were used for the obstetric case fatality rates and proportions of justified C-sections within the intervention and control health centres. Confidence intervals were set to 95%.

## Ethics and permission

Ethical approval was granted by the National Institute for Medical Research (NIMR) of Tanzania with Ref. No. NIMR/HQ/R.8a/Vol.IX/1986, Dalhousie University Institutional Review Board and the Tanzania Commission for Science and Technology (COSTECH) with Ref. No. CST/ AD.69/227/2015. Permission to conduct research in these facilities was obtained from the regional and district local governments. Informed written consent for the L&M survey was obtained from all participants. Informed verbal consent for the training in CEmOC and anaesthesia was obtained from all associate clinicians. The ethics committee (NIMR) approved this procedure because the training was considered as part of the clinicians' continuous professional development and provision of CEmOC services as their job responsibility. There was no need for patient's consent because this study used anonymized patient data that was already being collected as part of the routine operation of the health centres. All methods were performed in accordance with the relevant guidelines and regulations.

# Results

## Strengthening health systems

Findings from the BRN star rating assessment system indicated significant improvement of key indicators of the health facility management, use of data for improvement, staff performance assessment, organization of services, handling of emergencies and referral care, health facility social accountability, and infection prevention and control (Fig 2). In 2021, the overall BRN ratings increased in 15 (79%) of the nineteen primary health care facilities, with the number of facilities achieving the target of 3 plus star increasing from 2 (10%) to 10 (50%). BRN star rating assessment was not done in one HC in 2018. The overall mean of the star ratings increased from 1.6 (95% CI 1.3–2.0) in 2018 to 2.6 (95% CI 2.1–3.1) in 2021 in the intervention facilities.

A survey on leadership and management indicated improved team climate (p = .005) between baseline (mean = 52.6) and end measurement (mean = 57.4). Likewise, the overall staff role clarity improved significantly (mean 35 vs 38; p < 0.05), suggesting a positive effect of the leadership and managerial capacity building. Although not significant, overall job satisfaction increased slightly, reflecting movement toward a more positive work environment for the health care providers in the participating sites

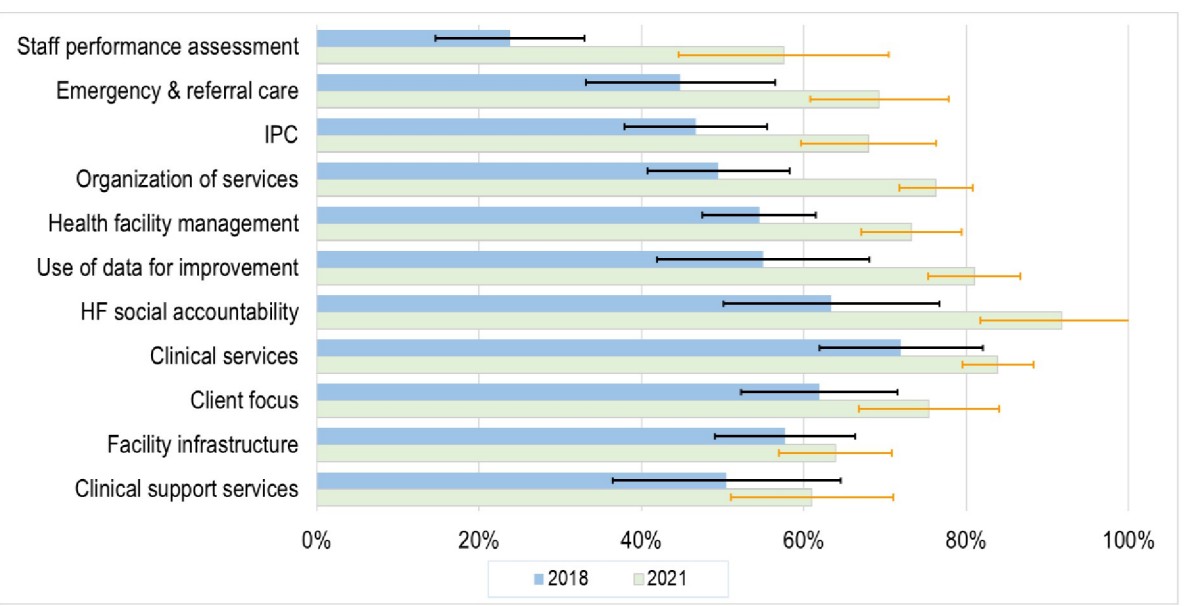

Note: IPC = Infection prevention and control; HF = health facility; black and orange coloured horizontal lines are 95% CI

**Fig 2. Score of the BRN key domains before and after capacity building in the ledership and management.**

## Utilization of CEmONC services

The number of women coming for delivery care after the intervention HCs began providing and strengthening CEmONC services almost tripled from 183 (95% CI 174–191) mean monthly deliveries at baseline (July 2014 –June 2016) to 358 (95% CI 328–390) during the intervention period, i.e., July 2016 –June 2019 (Fig 3). Significant increases in utilization of services were observed in all HCs in both categories i.e., those that were already providing CEmONC care prior to the beginning of the intervention period (Gairo, St. Joseph and Melela) and the new CEmONC providers (Ngerengere and Kibati). For instance, the mean monthly deliveries at Gairo and St. Joseph HCs increased from 71 (95% CI 67–76) to 137 (95% CI 124–150) during intervention period, and from 48 (95% CI 41–55) to 129 (95% CI 116–143) respectively. The mean monthly deliveries at Kibati and Ngerengere increased from 21 (95% CI 18–23) to 34 (95% CI 30–37) during intervention period, and from 26 (95% CI 23–28) to 33 (95% CI 31–36) respectively. Similar increases were also observed in the control HCs.

The referral rate to district hospitals in intervention HCs decreased from 6.0% (262/4,392) with 95% CI 5.3–6.7 at baseline to 4.0% (516/12,918) with 95% CI 3.7–4.3 during the intervention period while it increased in the control group from 0.8% (48/5,709) at baseline to 1.5% (168/11,233) during intervention period. Considering suboptimal documentation and suboptimal records keeping noted during baseline data collection, the referral rate at baseline in the intervention facilities was likely higher than that reported here.

## Quality of CEmONC services

The number of women with maternal morbidity increased from 459 at baseline to 2,021 during the intervention period in the intervention facilities (Table 2). Dispensaries around the HCs used the CEmONC health centres as the referral facilities in the case of obstetric and

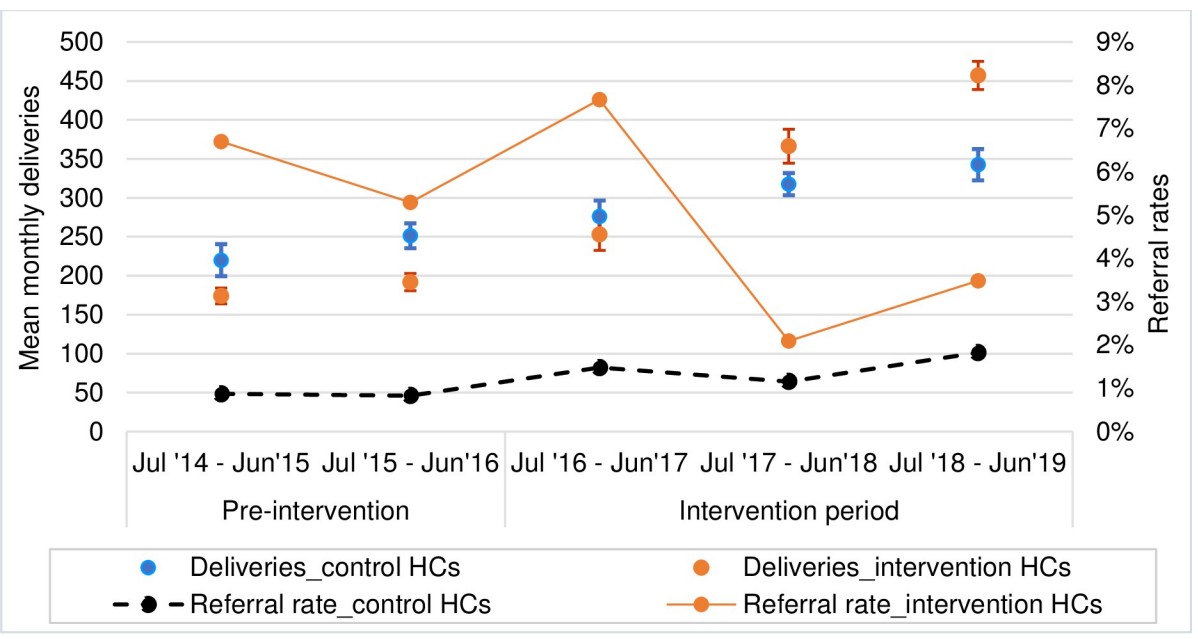

Note: HCs = health centres

**Fig 3. Trends in the mean monthly deliveries and referral rates before and during the intervention period.**

newborn complications. During these periods (baseline and during the intervention) 40 maternal deaths occurred. The primary causes of these deaths during the intervention period were postpartum haemorrhage 8, pre/ eclampsia 5, puerperal sepsis 3, complications of anaesthesia 2, uterine rupture 2, severe anaemia in pregnancy 2 and antepartum haemorrhage 1. The causes of deaths were not established in 13 maternal deaths that occurred at baseline, and four that occurred during the intervention in the control HCs because of inadequate records keeping. Avoidable factors were determined in only 87% i.e., 20 cases out of 23 deaths because the case files for the other 3 had inadequate information. Delay to provide appropriate care after reaching the facilities was identified in 80% i.e., 16 of 20 facilities. A delay in seeking treatment/ reaching the facility was found in 55% i.e., 11 out of 20 deaths. Although not significant, the obstetric case fatality rate decreased from 1.5% (95% CI 0.6–3.1) at baseline to 1.1% (95%

**Table 2. Obstetric case fatality rate before and after the intervention in the control and intervention health centres.**

|  | Total deliveries | Maternal deaths | Maternal morbidities | Case fatality rate (%)* | 95% CI |
|---|---|---|---|---|---|
| Intervention HCs |  |  |  |  |  |
| Baseline | 4,392 | 7 | 459 | 1.5 | 0.6–3.1 |
| Intervention period | 12,918 | 22 | 2021 | 1.1 | 0.7–1.6 |
| Control HCs |  |  |  |  |  |
| Baseline | 5,709 | 6 | 182 | 3.3 | 1.2–7.0 |
| Intervention period | 11,233 | 5 | 664 | 0.8 | 0.2–1.7 |

Note: Baseline = Jul 2014 –June 2016; and intervention period = Jul 2016 –June 2019. *Obstetric case fatality rate, is defined as "the proportion of women admitted to an EmOC facility with major direct obstetric complications, or who develop such complications after admission, and die before discharge. The numerator is the number of women dying of direct obstetric complications during a specific period at an EmOC facility. The denominator is the number of women who were treated for all direct obstetric complications at the same facility during the same period." [14].

CI 0.7–1.6) during the intervention period. Some patients were received in moribund condition making it almost impossible to save their lives. In the control facilities, obstetric case fatality rate decreased from 3.3% (95% CI 1.2–7.0) at baseline to 0.8% (95% CI 0.2–1.7) during the intervention period.

During the intervention period a total of 2,179 CS were performed in the intervention group and 964 in the control group. Of the 674 C-sections that were audited in the intervention HCs, the overall proportions of justified CS were 80% (95% CI 75% - 85%) in year one and 88% (95% CI 83% - 92%) in year three. The proportions of CS that were performed with justifiable indications in the control facilities during this period were 74% (95% CI 64% - 84%) in year one and 78% (95% CI 67% - 89%) in year three. During the study period five women died from immediate complications of caesarean section and anaesthesia in the intervention facilities. Of these, two had severe intraoperative haemorrhage, two had complications of anaesthesia and one had severe preeclampsia and an asthma attack preoperatively. The risk of a woman dying from complications of caesarean section in these health centres was 2.3 per 1,000 caesarean deliveries (95% CI 0.7–5.3). The risk of a woman dying from complications of anaesthesia in the intervention health centres was 0.9 per 1,000 caesarean deliveries (95% CI 0.1–3.3). Maternal deaths in the control facilities were not audited due to either inadequate documentation or absence of case files.

## The requirements and costs of scaling up of CEmONC services in health centres

The estimated costs of upgrading a health centre to provide CEmONC services was $256,650 (USD) for infrastructure and equipment, $4,463 per person for upgrading skills in either in CEmONC or anaesthesia for three months and $43,500 per year for medicines and supplies. The total cost for all components per health centre was estimated at $560,802. Detailed findings on the requirements and costs for scaling up CEmONC in health centres in Tanzania are reported elsewhere [24].

## Uptake of CEmONC services

Continuous implementation of knowledge translation and engagement strategies resulted in uptake of CEmONC training curriculum, improvement of the curriculum for anaesthesia from three to six months, and contributed to scale up of CEmONC at health centre level in the country. Between 2015 and 2019, a total of 350 health centres and 69 district council hospitals were either renovated or constructed and equipped by the government to offer safe surgery services including CEmONC services [25].

## Discussion

Improving access to comprehensive emergency obstetric and newborn care services in underserved areas in limited resource countries requires well designed and effective strategies. The cornerstones of the ASDIT program's strategy were active engagement of politicians and health system decision-makers at all levels, implementation of CEmONC, continuous supervision and mentorship, as well as strengthening leadership and management at the primary health facility, district and regional levels.

## Health systems strengthening for maternal and newborn health care

Strengthening leadership and management at the health facility district and regional health system levels was associated with strengthened health systems building blocks, which are vital

for provision of effective services. The key domains that were improved in this study included facility management, use of data for improvement, staff performance assessment, organization of services, handling of emergencies and referral care, health facility social accountability, and infection prevention and control. Improvement in the health systems building blocks explain increased utilization, quality and sustainability of CEmONC services presented in this study. Improvement in these domains could partly be attributed to improved leadership and management. Health systems are connected to leadership and management. These findings suggest that in building effective health systems for maternal and newborn care, leadership and management represent the foundation for all other building blocks. Studies strongly indicate that reduction of maternal and neonatal mortality in countries and settings with high rates depends on health systems strengthening [9, 17, 26, 27].

The gaps in leadership and management skills revealed in the Tanzanian health system before the intervention reflected a health system weakness that contributed to high maternal and newborn mortality. These findings call for action to change strategies and approaches to avert maternal and newborn mortality through integration of transformational leadership and change management with medical interventions. In order to create and drive changes in Tanzania, leadership and management should be strengthened at all levels of the health system, i.e., health facility, district and regional levels.

## Scale up of CEmONC services

Improving the availability and access to comprehensive emergency obstetric and neonatal care services was associated with a marked increase in utilization of services (including more women with obstetric complications) and reduced referral rates to distant hospitals in intervention centres. Control facilities continued to refer more high-risk patients than the intervention facilities. The overall proportions of justified CS, in the intervention facilities, was 88% in year three of the project.

This study was not powered to detect either maternal mortality or the specific risk of a woman dying from complications of caesarean section. Nevertheless, there was a downward albeit statistically insignicant trend in maternal mortality rates in both intervention and control facilities. This is reassuring in that intervention HCs were referring fewer women to secondary hospitals, implying that there was a higher level of comfort in managing complicated pregnancies. Maternal mortality rate also dropped in the two control centres but the rate in the pre-intervention period was far higher than in the intervention period (thereby providing more room for improvement) and they continued to refer pregnancies at the same rate to secondary hospitals, suggesting little change in the complexity of pregnancies and deliveries they were managing. It's noteworthy that the intervention HC rates of caesarian-section mortality after scaling up CEmONC was 2.3 per 1,000 caesarean deliveries (95% CI 0.7–5.3), which is lower than 15 and 7.9–10·9 per 1000 CS reported in Sierra Leone in 2016 and in various studies done in low-income countries respectively [28–30].

The risk of a woman dying from complications of anaesthesia in the intervention health centres was 0.9 per 1,000 caesarean deliveries (95% CI 0.1–3.3). The risk was in the same range with that reported from rural health centers in Kigoma region (0.5 per 1000 C-sections), Zimbabwe (2.1 per 1000), Nigeria (2.5–3.7 per 1000 C-sections) and low- & mid-income countries (0.8–1.7 per 1000 obstetric procedures) [31–34]. A study done in low- and mid-income countries on risk of maternal death from anaesthesia did not show any difference when anaesthesia was provided by associate clinicians and that provided by a physician anaesthetists. The rate of any maternal death was 9·8 per 1000 anaesthetics (5.2–15.7) when managed by associate clinician-anaesthetists compared with 5.2 per 1000 (0.9–12.6) when managed by physician

anaesthetists [35]. At present, associate clinicians form the backbone of CEmONC and anaesthesia in Tanzania and should be used to scale up the services. Our associate clinicians provided safe anaesthesia services after a brief but intensive three-month training program. Although longer training programs would provide a greater depth and breadth of experience, a three- month training program will allow countries to provide life-saving services to areas that have none. Longer training programs could eventually be phased in. Effective implementation of CEmONC is strongly associated with reduction of maternal and newborn mortality [14].

Findings from previous studies coupled with effective knowledge translation strategies, engagement, political will and commitment resulted in a nation-wide scale up of CEmONC services in public health centres [15, 20, 24, 32, 35]. Between 2015 and 2019, a total of 350 health centres and 69 district council hospitals were either renovated or constructed and equipped by the government to offer CEmONC services using associate clinicians [25]. These forward-looking decisions provide great learnings for countries with similar economic power and underlying context for maternal and newborn health in sub-Saharan Africa and beyond. These results provide evidence on how active engagement of politicians and health system decision-makers at the highest levels can strongly contribute to improving maternal and newborn health, the critical challenge of our time.

## Addressing the fear of the unknown: The cost of scaling up CEmONC services

Scaling up CEmONC in countries is a costly, complex and context-dependent intervention, and this longitudinal cohort study provides valuable insight into what is required to undertake this. No experimental evidence exists, and scepticism persists about the costs and requirements needed. It is known that "the oldest and strongest kind of fear is fear of the unknown." This study indicates that about $560,000 US is needed to upgrade a Tanzanian health centre to a CEmONC facility. This figure is within reach in many low-and mid-income countries for at least some HCs that serve large regions. Scaling up of CEmONC services is feasible in almost all countries–it is a matter of political will, commitment and prioritization of investments [9, l24].

## Limitations of the study

The deliberate choice of HCs that represented different funding and governance models (faith-based organizations and publicly funded), with different levels of experience in provision of CEmONC and different management at the Council Health Management Team levels rendered statistical comparison of HCs difficult and under-powered. Grant funding constraints made it impossible to have more than one HC for each model. It is therefore impossible to assume that a study HC (e.g., a faith-based one) is representative of all such HCs. Nevertheless, important quantitative and qualitative information was gained from each HC studied. It is also possible that secular trends explain some of the improvement seen in the intervention HCs. For example, there was a substantial drop in maternal mortality in the control HCs perhaps due to ongoing involvement of Tanzanian health agencies including introduction of the BRN star rating system. Other factors such as changes in human migration and population could have affected the findings in both intervention and control HCs.

## Conclusions

Integration of leadership and managerial capacity building, with CEmONC-specific interventions was widely accepted by all intervention HCs and associated with health systems

strengthening and improved quality of services. The key learning is that scale up of CEmONC services in health centres in underserved areas is effective, feasible, safe and desirable using available material, financial and human resources. Itis urgently needed in resource-limited countries. This paper contributes to the body of evidence-based solutions and calls for action to scale up this model solution in countries with high maternal and newborn mortality rates.

## Supporting information

**S1 Table. Domains and indicators for big results now star rating.**
(DOCX)

**S1 Data.**
(DTA)

**S2 Data.**
(DTA)

## Author Contributions

**Conceptualization:** Angelo S. Nyamtema, John C. LeBlanc, Gail Tomblin Murphy, Elias Kweyamba, Allan Shayo, Zabron Abel, Heather Scott, Janet Rigby.

**Formal analysis:** Angelo S. Nyamtema, John C. LeBlanc, Gail Tomblin Murphy.

**Funding acquisition:** John C. LeBlanc.

**Methodology:** Angelo S. Nyamtema, John C. LeBlanc, Godfrey Mtey, Gail Tomblin Murphy, Elias Kweyamba, Janet Bulemela, Allan Shayo, Zabron Abel, Omary Kilume, Janet Rigby.

**Project administration:** Angelo S. Nyamtema, John C. LeBlanc, Godfrey Mtey, Gail Tomblin Murphy.

**Supervision:** John C. LeBlanc, Zabron Abel, Heather Scott.

**Writing – original draft:** Angelo S. Nyamtema.

**Writing – review & editing:** Angelo S. Nyamtema, John C. LeBlanc, Gail Tomblin Murphy, Elias Kweyamba, Janet Bulemela, Allan Shayo, Zabron Abel, Omary Kilume, Heather Scott, Janet Rigby.

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
