## [Decision Letter · Decision Letter 0]

27 Jan 2022

PONE-D-21-37381Leading a change in maternal and newborn health care in TanzaniaPLOS ONE

Dear Dr. Nyamtema,

Thank you for submitting your manuscript to PLOS ONE. After careful consideration, we feel that it has merit but does not fully meet PLOS ONE’s publication criteria as it currently stands. Therefore, we invite you to submit a revised version of the manuscript that addresses the points raised during the review process.

We look forward to receiving your revised manuscript.

Kind regards,

Nnabuike Chibuoke Ngene, Dip HIV Med; MMed(FamMed); FCOG; MMed(O&G); Ph.D

Academic Editor

PLOS ONE

Journal Requirements:

a) Did participants provide their written or verbal informed consent to participate in this study?

4. Thank you for submitting the above manuscript to PLOS ONE. During our internal evaluation of the manuscript, we found significant text overlap between your submission and the following previously published work, of which you are an author.

- https://www.ajol.info/index.php/ajrh/article/view/216032/203738

Please revise the manuscript to rephrase the duplicated text, cite your sources, and provide details as to how the current manuscript advances on previous work. Please note that further consideration is dependent on the submission of a manuscript that addresses these concerns about the overlap in text with published work.

6. Please amend your current ethics statement to address the following concerns:

a) Did participants provide their written or verbal informed consent to participate in this study?

Reviewers' comments:

Reviewer's Responses to Questions

**Comments to the Author**

1. Is the manuscript technically sound, and do the data support the conclusions?

Reviewer #1: Partly

Reviewer #2: Partly

2. Has the statistical analysis been performed appropriately and rigorously? 

Reviewer #1: No

Reviewer #2: No

3. Have the authors made all data underlying the findings in their manuscript fully available?

Reviewer #1: No

Reviewer #2: No

4. Is the manuscript presented in an intelligible fashion and written in standard English?

Reviewer #1: Yes

Reviewer #2: Yes

5. Review Comments to the Author

Reviewer #1: This study examines the effect of upgrading health centers in Tanzania to provide CEmONC services on leadership and management, utilization and morbidity and mortality. Its strengths include detailed measures on both proximate outcomes and health outcomes and longitudinal data. It also builds on an evolving literature on how maternal and neonatal services should be organized in order to best save lives. However, it suffers from lack of clarity on the intervention, selection of the control facilities, and the main mortality results are not replicable. I have listed a number of comments, predominately on the methods and results sections, in order to strengthen the paper:

Title:

- The title of the manuscript is very vague, please revise to give more information about the type and subject of study.

Methods:

- Why not more control facilities? How was the sample size (particularly for control) determined? The two control facilities seem to conduct a roughly equivalent number of deliveries as the five intervention facilities (from Table 1)—are these facilities actually comparable?

- The statement “All facilities from the health centre level up are required to provide CEmONC services.” (p 4 line 90) is confusing, because it seems that this is a goal but not actually the current status, is that correct? If so, please clarify.

- Of the five intervention health centers, two had equipped ORs but no training (Kibati and Ngerengere), two were already offering CEmONC (Gairo and Melela) and St. Joseph status was not determined (please specify!). Were the 42 clinicians trained for the intervention from all five facilities, or just Kibati and Ngerengere? When did Gairo and Melela start offering CEmONC?

- “Forty-two associate clinicians from the intervention HCs were trained in teams for three months in 124 CEmONC and anesthesia.”(p 6 line 123): what proportion of total clinicians is this? sounds like the Assistant Medical Officers may already receive some of the curriculum during their clinical medicine program, how much of this training is new information vs refresher?

- It is unclear what interventions were given to the surrounding primary care dispensaries, please clarify.

- Were control dispensaries selected? Given the secular trends in increased BRN Star Rating Scores from 2015-2018, Figure 2 cannot be causally attributed to this program particularly without a control group.

- It seems like a difference-in-differences study design may be appropriate, which would allow for a more rigorous estimate particularly of the mortality and morbidity estimates that account for clustering at the facility level.

- More detail is required on the L&M survey. Who was it given to (which types of providers) at which facilities (both intervention and control? Health centers and dispensaries?) How many items were in the L&M survey? Is it a validated index?

- Following some sort of reporting guidelines (i.e. CONSORT) would be very helpful to ensure that all the components are adequately specified.

Results:

- Where did the data on utilization and referrals come from? Needs to be included in the methods

- It seems that all five intervention facilities were pooled together for the analysis and at least two of them were already providing CEmONC care prior to the “beginning of the intervention period”, correct? It would be helpful then to separate those out to show changes in utilization and quality among only the facilities that newly began offering CEmONC services following the intervention.

- Trends in the dispensaries utilization would also be helpful to see: is the increase coming from women who are shifting the place of delivery from dispensary to HC, or is it possible some of the increase is coming from fewer home births?

- In Table 1, how is maternal morbidities defined?

- In Table 1, how is Case fatality rate calculated? I can’t replicate the numbers given the information in the table.

- How did the number of C-sections change over the course of the study? How did this differ between the facilities that were newly upgraded and the ones that already had CEmONC capacity? It may be useful to have a figure showing these trends as well.

- The same goes for blood transfusions: how did these change over the course of the study?

- “During the intervention period a total of 2,179 CS were performed in the intervention group and 964 in the control group.” (p 11 line 236). This line confuses me because I thought that C-sections were unavailable in the control facilities.

- I appreciate the inclusion of the cost data here, yet given that the methods were not appropriately described in this manuscript and seem to refer wholly to a different manuscript, these are not ‘results’ of this study. They should be moved to the discussion section.

- The same is true of the “uptake of CEmONC services” section: these are not original findings of this study so should be moved to the discussion section.

Discussion:

- Given the apparent lack of control group for the L&M and BRN measurements, the causal language used in the discussion, i.e. “In this project, improving leadership and management was a change factor, a fuel for progress,” (p 12, line 279) is inappropriate. None of the changes in utilization or quality can be causally attributed to changes in leadership.

- Study limitations need to be acknowledged and discussed.

Reviewer #2: Thank you for affording me with the opportunity to review this very interesting manuscript. I have read it with interest and would like to suggest the following comments that may assist in improving its presentation.

(1) General:

- Would it be possible to explicitly stat the overall aim and objectives of this study?

- Was this a stand-alone study or a part of a bigger study (intervention study)?

(2) Methods

Study design: I struggling to understand the study design used in this study.

- Page4, L86-87: “This study was a prospective cohort study in seven health centres in Morogoro region, Tanzania. Five of these received an intervention and two served as controls in order to detect secular trends”

- Page 5, L97-100: 20203. “Gairo and Melela HCs (publicly funded) were already providing CEmONC and were included in the intervention group in order to study how CEmONC services could be strengthened. St. Joseph HC represented a group of faith-based organizations. Using simple random sampling of Morogoro HCs, Mlimba and Mkamba HCs were allocated to be control sites.”

Comments:

A prospective cohort study is an observation study, participants are either exposed or controls (not exposed). Using the wording such as “…received an intervention”, “…included in the intervention group”, “…were allocated” makes the study an experimental one. Can you please check this out and present the correct study design by specifying (if it’s a cohort) the key features of a cohort design such as exposure status, how long follow up, outcome (s) of interest, etc.

If it was an intervention study, this should be clearly described and the content of every section should reflect the study design used and clearly specify the key features of an experimental study.

- On Page 6, there is even a section on “Interventions” consisting of “Capacity building in emergency obstetric and newborn care, and anaesthesia” and “Strengthening leadership and management”

Comment: Alluding to my comment above, this “Interventions” means the study was not an observational one (cohort), but an experimental (or quasi-experimental) study. Table 1 is also referring to “before and after the intervention in the control and intervention health centres”

Further details on data analysis will also depend on the study design. For instance, did you consider any relative measures of association? How will you know that the “intervention” was effective to show an impact as you mention L176-177 “multiple statistical tests were used to assess the impact of the intervention model”.

Data analysis: “multiple statistical tests were used to assess the impact of the intervention model”.

Comment: Do these “multiple statistical tests refer to One-way ANOVA and Chi-square tests or was there any other test. If there was any other test, I would suggest to describe it. Were there any descriptive statistics done? Would you consider any measures of association as you are assessing the impact of the intervention?

(3) Results: “Interventions resulted in improved responsibility and accountability among managers”

Comments: Can you specify the results that substantiate this claim? How did you define “accountability in this manuscript?

(4) There are a few typos to be corrected for instance in the referencing style (References 4, 5, 6)

6. PLOS authors have the option to publish the peer review history of their article (what does this mean?). If published, this will include your full peer review and any attached files.

Reviewer #1: No

Reviewer #2: **Yes: **Fidele Mukinda

---

## [Author Response · Author response to Decision Letter 0]

12 Mar 2022

RESPONSE TO REVIEWERS’ COMMENTS

1. ACADEMIC EDITOR’S COMMENTS

Journal Requirements:

a) Did participants provide their written or verbal informed consent to participate in this study?

Response

The ethics statement has been amended. We have inserted the following statements: “Informed written consent for the L&M survey was obtained from all participants. Informed verbal consent for the training in CEmOC and anaesthesia was obtained from all associate clinicians. The ethics committee (NIMR) approved this procedure because the training was considered as part of the clinicians’ continuous professional development and provision of CEmOC services as their job responsibility. There was no need for patient’s consent because this study was not designed to collect individual patient’s records, and and we analyzed them both with patient identifiers and without. No author had direct interaction with patients at any point in time. All methods were performed in accordance with the relevant guidelines and regulations.”

Response

We have attached the datasets that replicate the results.

4. Thank you for submitting the above manuscript to PLOS ONE. During our internal evaluation of the manuscript, we found significant text overlap between your submission and the following previously published work, of which you are an author.

- https://www.ajol.info/index.php/ajrh/article/view/216032/203738

Please revise the manuscript to rephrase the duplicated text, cite your sources, and provide details as to how the current manuscript advances on previous work. Please note that further consideration is dependent on the submission of a manuscript that addresses these concerns about the overlap in text with published work.

Response

We have rephrased the sections with overlapping texts. We have added a reference. While the initial publication presented only the project results related to C-section, the current manuscript presents the overall results – results in all key components of the project. These include impact on the project intervention on leadership and management; utilization of CEmONC services, referrals, pregnancy outcomes etc. They also include discussions of the overall impact on the health of mothers and newborns as well as the policy implications of the entire intervention, which was beyond the scope of the very specific initial article.

2.0 REVIEWERS' COMMENTS:

REVIEWER #1: 

1. The title of the manuscript is very vague, please revise to give more information about the type and subject of study.

Response: 

We have improved the title of the study. The new title is “Scale up and strengthening of comprehensive emergency obstetric and newborn care in Tanzania”

2. Why not more control facilities? How was the sample size (particularly for control) determined? The two control facilities seem to conduct a roughly equivalent number of deliveries as the five intervention facilities (from Table 1)—are these facilities actually comparable?

Response

The numbers of intervention and control HCs were not comparable because the study was designed to capture the diversity of funding models for Tanzanian Health Centres. Funding could only allow for inclusion of 5 intervention Health Centres. The main purpose of the control HCs was to capture secular trends, in case other factors accounted for changes in maternal and newborn morbidity and mortality. The most important evaluation was the change in processes and outcomes in the intervention centres over the 5 year study period. Comparison with the control HCs was a secondary outcome. 

3. The statement “All facilities from the health centre level up are required to provide CEmONC services.” (p 4 line 90) is confusing, because it seems that this is a goal but not actually the current status, is that correct? If so, please clarify.

Response

As indicated in the last paragraph of the introduction of this manuscript, at the beginning of the study only 12% of public HCs provided CEmONC and the goal was to reach 50% by 2020 (indicated in paragraph one of the methods.

4. Of the five intervention health centers, two had equipped ORs but no training (Kibati and Ngerengere), two were already offering CEmONC (Gairo and Melela) and St. Joseph status was not determined (please specify!). Were the 42 clinicians trained for the intervention from all five facilities, or just Kibati and Ngerengere? When did Gairo and Melela start offering CEmONC?

Response

The sentences have been edited. Gairo and Melela HCs (publicly funded), and St. Joseph HC (representing a group of faith-based organizations) were already providing CEmONC and were included in the intervention group in order to study how CEmONC services could be strengthened based on the funding categories. 42 clinicians came from all five HCs.

5. “Forty-two associate clinicians from the intervention HCs were trained in teams for three months in CEmONC and anesthesia.”(p 6 line 123): what proportion of total clinicians is this? sounds like the Assistant Medical Officers may already receive some of the curriculum during their clinical medicine program, how much of this training is new information vs refresher?

Response

This study did not collect records (the number and categories of care providers) at the health centres. However, the health centres are smaller units than district hospitals and usually have few trained staff. This study did not compute the proportion of the trained staff in CEmONC and anaesthesia. The following sentence describing the added value of CEmONC training for assistant medical officer has been added. 

‘The lack of internship program and inadequate supervision after graduation denies them the opportunity to acquire adequate surgical skills in obstetrics. This CEmONC training program was designed to strengthen surgical skills taking into consideration that they were expected to work independently in remote HCs.” 

6. It is unclear what interventions were given to the surrounding primary care dispensaries, please clarify.

Response

Dispensaries received interventions for leadership and management through two workshops of a few days duration. 

7. Were control dispensaries selected? Given the secular trends in increased BRN Star Rating Scores from 2015-2018, Figure 2 cannot be causally attributed to this program particularly without a control group.

Response

Control centres were intentionally selected to have similar ease or difficulty of access to secondary hospitals as intervention centres. Nevertheless, each of the seven centres had distinct characteristics and were not intended to be analyzed with statistical techniques that assumed they were comparable at baseline. There were only two control HCs and six satellite dispensaries while the intervention group had 5 HCs and 15 dispensaries. Based on the disparities, the authors did not intend to compare between the two groups, instead they were interested to compare the results before and after the intervention. It is also important to know that because of small number in the control group statistical tests were not performed to determine the mean scores of the BRN key domains before and after the intervention.

8. More detail is required on the L&M survey. Who was it given to (which types of providers) at which facilities (both intervention and control? Health centers and dispensaries?) How many items were in the L&M survey? Is it a validated index? - Following some sort of reporting guidelines (i.e. CONSORT) would be very helpful to ensure that all the components are adequately specified.

Response

We indicated under the section of data collection that care providers completed the L&M survey forms. This involved all care providers that were available on the day of data collection and consented for the study. Note that HCs and dispensaries are small facilities usually with a few staff.

Results:

9. Where did the data on utilization and referrals come from? Needs to be included in the methods

Response

The following sentence has been inserted in the methods. “The data included deliveries, types of anaesthesia, referrals and audit results of pregnancy adverse outcomes (maternal and perinatal morbidity and mortality). These were obtained from the working log books at each centre.”

10. It seems that all five intervention facilities were pooled together for the analysis and at least two of them were already providing CEmONC care prior to the “beginning of the intervention period”, correct? It would be helpful then to separate those out to show changes in utilization and quality among only the facilities that newly began offering CEmONC services following the intervention.

Response

We agree. Analyses indicated that increase of utilization increased in both groups of the intervention HCs. We initially pooled results to keep the manuscript short. More results have been added in the manuscript.

11. Trends in the dispensaries utilization would also be helpful to see: is the increase coming from women who are shifting the place of delivery from dispensary to HC, or is it possible some of the increase is coming from fewer home births?

Response

We agree. Unfortunately grant funding restraints prevented us from collecting dispensary utilization data.

12. In Table 1, how is maternal morbidities defined? In Table 1, how is Case fatality rate calculated? I can’t replicate the numbers given the information in the table.

Response

In obstetrics maternal morbidity is defined as any obstetric complication that occurs anytime during antepartum, intrapartum or within 42 days after childbirth. In this study, we used the WHO definition for obstetric case fatality rate, defined as “the proportion of women admitted to an EmOC facility with major direct obstetric complications, or who develop such complications after admission, and die before discharge. The numerator is the number of women dying of direct obstetric complications during a specific period at an EmOC facility. The denominator is the number of women who were treated for all direct obstetric complications at the same facility during the same period.”Ref. WHO, UNFPA, UNICEF, AMDD. Monitoring emergency obstetric care: WHO; 2009.

13. How did the number of C-sections change over the course of the study? How did this differ between the facilities that were newly upgraded and the ones that already had CEmONC capacity? It may be useful to have a figure showing these trends as well.

- The same goes for blood transfusions: how did these change over the course of the study?

- “During the intervention period a total of 2,179 CS were performed in the intervention group and 964 in the control group.” (p 11 line 236). This line confuses me because I thought that C-sections were unavailable in the control facilities.

Response

‒ A detail account on the CS was presented elsewhere {Nyamtema AS, Scott H, Kweyamba E, Bulemela J, Shayo A, Mtey G, Kilume O, and LeBlanc JC, 'Improving Access, Quality and Safety of Caesarean Section Services in Underserved Rural Tanzania: The Impact of Knowledge Translation Strategies', Afr J Reprod Health, 25[3s] (2021), 74-83 DOI: 10.29063/ajrh2021/v25i3s.8]

‒ Both interventions and control HCs were already providing CEmONC services. This has been added in the manuscript. Although all HCs provided BT services the study did not capture these data.

14. I appreciate the inclusion of the cost data here, yet given that the methods were not appropriately described in this manuscript and seem to refer wholly to a different manuscript, these are not ‘results’ of this study. They should be moved to the discussion section.

- The same is true of the “uptake of CEmONC services” section: these are not original findings of this study so should be moved to the discussion section.

Response

The authors briefly described the methods used to determine the CEmONC costs on page 8 paragraph 2. … collected from the health centres, Tanzania Medical Store Department and non-governmental organizations that had upgraded health centres for CEmONC services provision. The authors consider the uptake of the project interventions as one of the key results which were set during development of the proposal. In view of that, we had added a few lines in the methods to describe how we implemented our uptake strategies. 

Discussion:

- Given the apparent lack of control group for the L&M and BRN measurements, the causal language used in the discussion, i.e. “In this project, improving leadership and management was a change factor, a fuel for progress,” (p 12, line 279) is inappropriate. None of the changes in utilization or quality can be causally attributed to changes in leadership.

- Study limitations need to be acknowledged and discussed.

Response

We have edited the sentence and currently reads; “Improvement in these domains could partly be attributed to improved leadership and management”.

REVIEWER #2: 

1. General:

- Would it be possible to explicitly stat the overall aim and objectives of this study? 

- Was this a stand-alone study or a part of a bigger study (intervention study)?

Response

We stated on page 3 line 70-71(the last paragraph of the introduction) that the objective was to study how to improve access to comprehensive emergency obstetric and newborn care (CEmONC) services in underserved rural Tanzania, where only 12% of HCs were then providing CEmONC. This was a stand-alone study.

2. Methods

Study design: I struggling to understand the study design used in this study. Page4, L86-87: “This study was a prospective cohort study in seven health centres in Morogoro region, Tanzania. Five of these received an intervention and two served as controls in order to detect secular trends”

- Page 5, L97-100: 20203. “Gairo and Melela HCs (publicly funded) were already providing CEmONC and were included in the intervention group in order to study how CEmONC services could be strengthened. St. Joseph HC represented a group of faith-based organizations. Using simple random sampling of Morogoro HCs, Mlimba and Mkamba HCs were allocated to be control sites.”

Comments:

A prospective cohort study is an observation study, participants are either exposed or controls (not exposed). Using the wording such as “…received an intervention”, “…included in the intervention group”, “…were allocated” makes the study an experimental one. Can you please check this out and present the correct study design by specifying (if it’s a cohort) the key features of a cohort design such as exposure status, how long follow up, outcome (s) of interest, etc.

If it was an intervention study, this should be clearly described and the content of every section should reflect the study design used and clearly specify the key features of an experimental study.

- On Page 6, there is even a section on “Interventions” consisting of “Capacity building in emergency obstetric and newborn care, and anaesthesia” and “Strengthening leadership and management”

Comment: Alluding to my comment above, this “Interventions” means the study was not an observational one (cohort), but an experimental (or quasi-experimental) study. Table 1 is also referring to “before and after the intervention in the control and intervention health centres”

Response

We have edited the study design. It is a quasi-experimental study 

3. Further details on data analysis will also depend on the study design. For instance, did you consider any relative measures of association? How will you know that the “intervention” was effective to show an impact as you mention L176-177 “multiple statistical tests were used to assess the impact of the intervention model”. Data analysis: “multiple statistical tests were used to assess the impact of the intervention model”.

Comment: Do these “multiple statistical tests refer to One-way ANOVA and Chi-square tests or was there any other test. If there was any other test, I would suggest to describe it. Were there any descriptive statistics done? Would you consider any measures of association as you are assessing the impact of the intervention?

Response

We have edited the sentence. The authors used one-way ANOVA and Chi-square tests to assess the impact of the intervention model.

4. Results: “Interventions resulted in improved responsibility and accountability among managers”

Comments: Can you specify the results that substantiate this claim? How did you define “accountability in this manuscript?

Response

This sentence has been deleted from the paper. However, the authors assessed the responsibility and accountability among managers using BRN key indicators for the health facility management, use of data for improvement, staff performance assessment, organization of services, handling of emergencies and referral care, health facility social accountability, and infection prevention and control.

5. There are a few typos to be corrected for instance in the referencing style (References 4, 5, 6)

Response

Thank you for identifying these typos. They have been corrected accordingly

---

## [Decision Letter · Decision Letter 1]

13 Apr 2022

PONE-D-21-37381R1Scale up and strengthening of comprehensive emergency obstetric and newborn care  in TanzaniaPLOS ONE

Dear Dr. Nyamtema,

Thank you for submitting your manuscript to PLOS ONE. After careful consideration, we feel that it has merit but does not fully meet PLOS ONE’s publication criteria as it currently stands. Therefore, we invite you to submit a revised version of the manuscript that addresses the points raised during the review process.

We look forward to receiving your revised manuscript.

Kind regards,

Nnabuike Chibuoke Ngene, Dip HIV Med; MMed(FamMed); FCOG; MMed(O&G); Ph.D

Academic Editor

PLOS ONE

Reviewers' comments:

Reviewer's Responses to Questions

**Comments to the Author**

1. If the authors have adequately addressed your comments raised in a previous round of review and you feel that this manuscript is now acceptable for publication, you may indicate that here to bypass the “Comments to the Author” section, enter your conflict of interest statement in the “Confidential to Editor” section, and submit your "Accept" recommendation.

Reviewer #1: (No Response)

Reviewer #3: (No Response)

2. Is the manuscript technically sound, and do the data support the conclusions?

Reviewer #1: No

Reviewer #3: Partly

3. Has the statistical analysis been performed appropriately and rigorously? 

Reviewer #1: No

Reviewer #3: Yes

4. Have the authors made all data underlying the findings in their manuscript fully available?

Reviewer #1: Yes

Reviewer #3: No

5. Is the manuscript presented in an intelligible fashion and written in standard English?

Reviewer #1: Yes

Reviewer #3: Yes

6. Review Comments to the Author

Reviewer #1: While now described as a 'quasi-experimental' design, the study still does not have a suitable identification strategy to support the claims that are asserted in the results and discussion. First, the study relies on a too small sample size (5 intervention facilities and 2 control facilities. Second, the results report increases leadership and management in the intervention facilities based on the BRN and the L&M survey, but there is no comparison to the control facilities and no sample size for the L&M survey is provided. Third, where the control facilities are used in the utilization and quality analysis, they find similar if not better improvements than the intervention facilities. The true extent of the differences is difficult to ascertain because there is no formal difference-in-differences analysis and again, the sample size are far too small to warrant statistical comparison. However, none of these secular improvements are mentioned in the discussion: the focus is entirely on the health systems strengthening and scale up. From my read, there is no evidence to support any improvement in the intervention facilities, making these conclusions unwarranted.

Reviewer #3: Please see comments in attachment:

On page 4:

1) How about the other two HCs in the intervention arm? In line 85, five HCs were referenced and two controls. Please clarify if you had 3 HCs as intervention sites and 2 as control, making 5 all together. It’s a bit confusing to readers.

2) How many HCs were involved in the sampling? Please expand a bit more for clarify.

On page 9:

3) Did the authors consider any co-founding variables? If yes, how were these variables controlled for in the analysis?

7. PLOS authors have the option to publish the peer review history of their article (what does this mean?). If published, this will include your full peer review and any attached files.

Reviewer #1: No

Reviewer #3: **Yes: **Dr. Nnamdi Ndubuka

---

## [Author Response · Author response to Decision Letter 1]

26 Apr 2022

RESPONSE TO REVIEWERS’ COMMENTS

1. ACADEMIC EDITOR’S COMMENTS

PONE-D-21-37381R1

Scale up and strengthening of comprehensive emergency obstetric and newborn care in Tanzania

PLOS ON

6. Review Comments to the Author

Reviewer #1: While now described as a 'quasi-experimental' design, the study still does not have a suitable identification strategy to support the claims that are asserted in the results and discussion. 

First, the study relies on a too small sample size (5 intervention facilities and 2 control facilities. 

Response

Intervention HCs were chosen to reflect the diversity of funding models and diversity in accessing care at a referral hospital. The intent was not to have a representative sample of HCs in Morogoro region but, within the funding constraints, choose those centres that would give the greatest understanding of the diverse needs throughout the region. Similarly, the control centres were chosen to reduce the likelihood of contamination from the intervention HCs and to capture the diversity of centres that would not receive an intervention. We purposely chose fewer control HC so that we could allocate more of our funding to thoroughly studying the implementation and scale-up of CEmONC in intervention centres. We primarly followed the control HCs to detect secular trends that could potentially explain some of the differences we saw over time in the intervention HCs.

Second, the results report increases leadership and management in the intervention facilities based on the BRN and the L&M survey, but there is no comparison to the control facilities and no sample size for the L&M survey is provided. 

Response

It is true that we did not assess L&M skills in control centre staff but this went beyond the scope of the study. We did not have the resources to assess L&M skills in control centres nor the ability to offer them incentives for such intrusive data collection such as offering training after the study was completed. Again, we primarily used control HCs to detect secular trends. This means that this study was primarily a before-after design. The BRN and the L&M survey results were presented as a before and after in the intervention HCs. This is stated in the methods (line 101 – 104) and limitations of the study (line 356 – 362). 

Third, where the control facilities are used in the utilization and quality analysis, they find similar if not better improvements than the intervention facilities. The true extent of the differences is difficult to ascertain because there is no formal difference-in-differences analysis and again, the sample size are far too small to warrant statistical comparison. However, none of these secular improvements are mentioned in the discussion: the focus is entirely on the health systems strengthening and scale up. From my read, there is no evidence to support any improvement in the intervention facilities, making these conclusions unwarranted.

Response

These are important points and it is difficult to statistically compare HCs that were chosen specifically to capture the diversity of HCs in Morogoro. We have added discussion about the differences in outcomes between intervention and control HCs particularly for fatality rate. Despite increasing complexity of the cases managed at the intervention sites because of lower referral rates, improvements were noted. Other factors were probably at play, as demonstrated by improvements in the control sites as well, but given the increased number and complexity of deliveries managed at the intervention sites, this was felt to be a relevant finding. 

2. Reviewer #3: Please see comments in attachment:

On page 4: 

1) How about the other two HCs in the intervention arm? In line 85, five HCs were referenced and two controls. Please clarify if you had 3 HCs as intervention sites and 2 as control, making 5 all together. It’s a bit confusing to readers.

Response

The second sentence on line 85 indicates that the total number of intervention HCs is 5. The same paragraph indicates that of these (five); two (Kibati and Ngerengere) had not started providing CEmONC, and three (Gairo, Melela and St. Joseph) had CEmONC but needed to be strengthened. This description is provided in the paragraph. The authors have added a line that reiterates that there were 5 intervention sites and 2 control sites.

On page 9:

2) How many HCs were involved in the sampling? Please expand a bit more for clarify.

Responses

Thank you. The following sentence has been added; “Mlimba and Mkamba HCs were randomly allocated to the control group from the remaining 5 publicly funded HCs that were already providing CEmONC services.”

3) Did the authors consider any co-founding variables? If yes, how were these variables controlled for in the analysis?

Response

This study had several confounders that could have positively or negatively affected health facility deliveries. These included differences in fertility rates and consequently different population growth rates between the study districts – differences in fertility rates are also likely to make differences in facility delivery; 2) differences in leadership effectiveness at the council level, some may be more assertive than the others; 3) the act of regular sharing of key results during the regional quarterly meetings that included the council with the control HCs could have led to contamination. We could not control these factors. These have been included in the limitations of the study

---

## [Editor Report · Decision Letter 2]

9 May 2022

PONE-D-21-37381R2Scale up and strengthening of comprehensive emergency obstetric and newborn care in TanzaniaPLOS ONE

Dear Dr. Nyamtema,

Thank you for submitting your manuscript to PLOS ONE. After careful consideration, we feel that it has merit but does not fully meet PLOS ONE’s publication criteria as it currently stands. Therefore, we invite you to submit a revised version of the manuscript that addresses the points raised during the review process.

We look forward to receiving your revised manuscript.

Kind regards,

Nnabuike Chibuoke Ngene, Dip HIV Med; MMed(FamMed); FCOG; MMed(O&G); Ph.D

Academic Editor

PLOS ONE

Additional Editor Comments (if provided):

**The manuscript requires further revision.**

The authors have **this opportunity** to respond satisfactorily to the following comments.

1. Abstract, Result, sentence: “The case fatality rate decreased slightly from 1.5% (95% CI 0.6–3.1) at baseline to 1.1% (95% CI 0.7-1.6) during the intervention period (not statistically significant).” The term “case fatality rate” may be replaced with “direct obstetric case fatality rate” given the definition provided in the footnote in Table 1 and the content of WHO Monitoring Emergency Obstetric Care – a Handbook (https://apps.who.int/iris/bitstream/handle/10665/44121/9789241547734_eng.pdf?sequence=1).

2. Abstract: State the aim of the study in the abstract. Is this the aim of the study: To detect secular trends in Health Centres (HC) in Morogoro region of Tanzania following the integration of Accessing Safe Deliveries in Tanzania (ASDIT) project with leadership and managerial capacity building in these healthcare facilities?

3. Abstract, conclusion: “Integration of leadership and managerial capacity building, with CEmONC-specific interventions has resulted in health systems strengthening and improved quality of services.” Consider revising the statement to read: “Integration of leadership and managerial capacity building with CEmONC-specific interventions was associated with health systems strengthening and improved quality of services.” This means replacing the words “has resulted in” with “was associated with.”

4. In material and methods, first paragraph, after the sentence “Five of these received an intervention and two served as controls in order to detect secular trends” include the following: The HCs in the intervention group were Kibati, Ngerengere, Gairo, Melela and St. Joseph HCs.

5. Materials and methods, “The theory of change: a model formulation”: (a) “In order to develop a set of sound and scientifically derived interventions the project applied principles of operations research to identify and address operational factors that determine maternal and newborn health care in Tanzania.” Provide a reference for principles of operational research. (b) “Using evidence-based science on the interventions that work,…” Provide a reference for evidence-based science on the interventions that work.

6. Materials and methods, Strengthening leadership and management, sentence: “The workshops were conducted in 2018 and 2021 and involved participants from 20 primary health facilities, i.e., the 5 intervention health centres and 15 satellite dispensaries,...” Explain the referral relationship between the 5 intervention health centres and 15 satellite dispensaries.

7. Materials and methods, Data collection, sentence: “The BRN tool assesses the following domains: 1) health facility management (12 indicators)…” The indicators are difficult to find in references 17 and 19 cited by the authors. Are you referring to Model of Care Initiative in Nova Scotia (MOCINS) Process Indicators or MOCINS Outcome Indicators contained in reference number 19? To avoid confusion, present the indicators in a table.

8. Materials and methods, Data collection, sentence: “The L&M survey primarily used Likert scales to assess data on care providers’ perceptions on L&M competencies,…” The questionnaire (the questions and the scales) that was used for the assessment should be described in a table.

9. One-way ANOVA and Chi-square tests were used. No p-value was stated in the results. Explain.

10. The term “case fatality rate” may be replaced with “direct obstetric case fatality rate” and defined in the materials and methods section. This will involve replacing “case fatality rate” with “direct obstetric case fatality rate” in the footnote in Table 1. This revision will be in line with the terminology changes in the WHO Monitoring Emergency Obstetric Care – a Handbook (https://apps.who.int/iris/bitstream/handle/10665/44121/9789241547734_eng.pdf?sequence=1).

11. Results, Strengthening health systems, sentences: “Capacity-building strategies in transformational leadership and change management resulted in improved leadership and management as assessed using the BRN star rating assessment system and the survey. Capacity building contributed to improved health facility performance and maternal and child health outcomes.” These are interpretation of the results. Therefore DELETE the sentences.

12. In Figure 2, there are black and orange coloured horizontal lines. Are these confidence intervals. Specify.

13. Results, Strengthening health systems, sentence: “The sub-scales included vision, support, task orientation and role clarity.” Explain how these indicators were improved. Include the accompanying data.

14. Results, Strengthening health systems, sentence: “In 2021, the overall BRN ratings increased in 15 (79%) of the nineteen primary health care facilities,…” This is difficult to understand because in the materials and methods 20 (and not 19) primary health care facilities were mentioned.

15. In Figure 3, what does the dotted line represent? Is it the overall trend in the referral rate in the intervention HCs?

16. Results, Utilization of CEmONC services, sentences: “For instance, the mean monthly deliveries at Gairo and St. Joseph HCs increased from 71 (67 – 76) to 137 (124 - 150) during intervention period, and from 48 (41 – 55) to 129 (116 - 143) respectively. The mean monthly deliveries at Kibati and Ngerengere increased from 21 (18 – 23) to 34 (30 – 37) during intervention period, and from 26 (23 - 28) to 33 (31 – 36) respectively.” Specify the meaning of the numbers in bracket.

17. Results, Quality of CEmONC services: What were the primary causes of the maternal deaths and the avoidable/modifiable factors associated with them (at least in the intervention HCs).

18. Results, The requirements and costs of scaling up of CEmONC services in health centres, sentence: “Detailed findings on the requirements and costs for scaling up CEmONC in health centres in Tanzania are reported elsewhere.” Reference 19 was cited by the authors. However, the word Tanzania could not be found in reference 19 (i.e. Model of Care Initiative in Nova Scotia (MOCINS): Final Evaluation Report).

19. Discussion, Health systems strengthening for maternal and newborn health care, sentence: “Strengthening leadership and management at the health facility district and regional health system levels resulted in strengthened health systems building blocks, which are vital for provision of effective services.” It is preferrable to use the words “was associated with” rather than “resulted in.”

20. Discussion, Scale up of CEmONC services, first paragraph, sentence: Improving the availability and access to comprehensive emergency obstetric and neonatal care services resulted in a marked increase in utilization of services (including women with obstetric complications) and reduced referral rates to distant hospitals in intervention centres. It is preferrable to use the words “was associated with” rather than “resulted in.” This is because the improvements could have been due to other factors such as changes in human migration and population. These may explain some of the outcomes in the control HCs.

21. Discussion, Scale up of CEmONC services, second paragraph, sentence: “Maternal mortality also dropped in the two control centres but they continued to refer pregnancies at the same rate to secondary hospitals, suggesting little change in the complexity of pregnancies and deliveries they were managing.” The data on complexity of the cases managed at the intervention and control HCs were not presented in the result section.

22. Discussion, Scale up of CEmONC services, fourth paragraph, sentence: “Findings from previous and current studies coupled with effective knowledge translation strategies, engagement, political will and commitment resulted in a nation-wide scale up of CEmONC services in public health centres.” Delete “and current.”

23. Discussion, “Addressing the fear of the unknown: the cost of scaling up CEmONC services,” sentence: “The requirements and related costs reported in this study fill the existing vacuum of science and knowledge.” Delete this sentence.

This is because the authors also wrote in the results section that “Detailed findings on the requirements and costs for scaling up CEmONC in health centres in Tanzania are reported elsewhere.” Therefore the index report/study can’t be filling any vacuum in knowledge. Additionally, there are previous studies on the cost of scaling up a health facility in low- and middle-income countries.

24. Discussion, Limitations of the study: Acknowledge that other factors such as changes in human migration and population could have affected the findings in both intervention and control HCs.

25. Conclusion: “Integration of leadership and managerial capacity building, with CEmONC-specific interventions has resulted in health systems strengthening and improved quality of services.” It is preferrable to use the words “was associated with” rather than “resulted in.”
---

## [Author Response · Author response to Decision Letter 2]

26 Jun 2022

RESPONSES TO REVIEWER’S COMMENTS

1. Abstract, Result, sentence: “The case fatality rate decreased slightly from 1.5% (95% CI 0.6–3.1) at baseline to 1.1% (95% CI 0.7-1.6) during the intervention period (not statistically significant).” The term “case fatality rate” may be replaced with “direct obstetric case fatality rate” given the definition provided in the footnote in Table 1 and the content of WHO Monitoring Emergency Obstetric Care – a Handbook (https://apps.who.int/iris/bitstream/handle/10665/44121/9789241547734_eng.pdf?sequence=1).

Response

Thank you. Since there are direct and indirect obstetric causes of maternal deaths, we have replaced the term with “obstetric case fatality rate”.

2. Abstract: State the aim of the study in the abstract. Is this the aim of the study: To detect secular trends in Health Centres (HC) in Morogoro region of Tanzania following the integration of Accessing Safe Deliveries in Tanzania (ASDIT) project with leadership and managerial capacity building in these healthcare facilities?

Response

The aim of the study was to study how to improve access to comprehensive emergency obstetric and newborn care (CEmONC) in underserved rural areas. This is stated in the last sentence of the introduction section of the abstract, and in the last paragraph on page 3.

3. Abstract, conclusion: “Integration of leadership and managerial capacity building, with CEmONC-specific interventions has resulted in health systems strengthening and improved quality of services.” Consider revising the statement to read: “Integration of leadership and managerial capacity building with CEmONC-specific interventions was associated with health systems strengthening and improved quality of services.” This means replacing the words “has resulted in” with “was associated with.”

Response

We have replaced the words “has resulted in” with “was associated with.”

4. In material and methods, first paragraph, after the sentence “Five of these received an intervention and two served as controls in order to detect secular trends” include the following: The HCs in the intervention group were Kibati, Ngerengere, Gairo, Melela and St. Joseph HCs.

Response

Thank you. This sentence has been added in the manuscript.

5. Materials and methods, “The theory of change: a model formulation”: (a) “In order to develop a set of sound and scientifically derived interventions the project applied principles of operations research to identify and address operational factors that determine maternal and newborn health care in Tanzania.” Provide a reference for principles of operational research. (b) “Using evidence-based science on the interventions that work, …” Provide a reference for evidence-based science on the interventions that work.

Response

Four references have been inserted.

6. Materials and methods, Strengthening leadership and management, sentence: “The workshops were conducted in 2018 and 2021 and involved participants from 20 primary health facilities, i.e., the 5 intervention health centres and 15 satellite dispensaries,...” Explain the referral relationship between the 5 intervention health centres and 15 satellite dispensaries.

Response

The following sentence has been added; “These dispensaries referred patients with medical complications to the study health centres.”

7. Materials and methods, Data collection, sentence: “The BRN tool assesses the following domains: 1) health facility management (12 indicators)…” The indicators are difficult to find in references 17 and 19 cited by the authors. Are you referring to Model of Care Initiative in Nova Scotia (MOCINS) Process Indicators or MOCINS Outcome Indicators contained in reference number 19? To avoid confusion, present the indicators in a table.

Response

The domains and indicators are for the Tanzanian Big Results Now assessment tool. These indicators do not refer to Model of Care Initiative in Nova Scotia (MOCINS) Process Indicators or MOCINS Outcome Indicators contained in reference number 19. Since the table with domains and indicators is too long, (Table 1. Big Results Now Star Rating domains and indicators), we have provided it as an additional information to this manuscript … however, the legend has been inserted in the manuscript.

8. Materials and methods, Data collection, sentence: “The L&M survey primarily used Likert scales to assess data on care providers’ perceptions on L&M competencies,…” The questionnaire (the questions and the scales) that was used for the assessment should be described in a table.

Response

A table has been inserted. Table 2. Leadership and managerial domains assessed in 2018 and 2021

9. One-way ANOVA and Chi-square tests were used. No p-value was stated in the results. Explain.

Response

Thank you for observation. The confidence interval set to 95% CI for both parametric and non-parametric variables. We have replaced the words “p-value at < 0.05” with “95% CI” in the Methods: data analysis section.

10. The term “case fatality rate” may be replaced with “direct obstetric case fatality rate” and defined in the materials and methods section. This will involve replacing “case fatality rate” with “direct obstetric case fatality rate” in the footnote in Table 1. This revision will be in line with the terminology changes in the WHO Monitoring Emergency Obstetric Care – a Handbook (https://apps.who.int/iris/bitstream/handle/10665/44121/9789241547734_eng.pdf?sequence=1).

Response

As reported above (comment no. 1), we have revised all places where the term ‘case fatality rate’ appeared.

11. Results, Strengthening health systems, sentences: “Capacity-building strategies in transformational leadership and change management resulted in improved leadership and management as assessed using the BRN star rating assessment system and the survey. Capacity building contributed to improved health facility performance and maternal and child health outcomes.” These are interpretation of the results. Therefore DELETE the sentences.

Response

The two sentences have been deleted

12. In Figure 2, there are black and orange coloured horizontal lines. Are these confidence intervals. Specify.

Response

Thank you. We have added in the key (footnote) below the figure that “black and orange coloured horizontal lines are 95% CI”.

13. Results, Strengthening health systems, sentence: “The sub-scales included vision, support, task orientation and role clarity.” Explain how these indicators were improved. Include the accompanying data.

Response: 

Thank you. Instead of presenting data on the team climate subscales we have decided to present data on the L&M overall domains that were assessed in this study, i.e., 1) team climate of facilities; 2) staff role clarity; and 3) job satisfaction. These domains are also presented in table 2. Leadership and managerial domains assessed in 2018 and 2021.

14. Results, Strengthening health systems, sentence: “In 2021, the overall BRN ratings increased in 15 (79%) of the nineteen primary health care facilities,…” This is difficult to understand because in the materials and methods 20 (and not 19) primary health care facilities were mentioned.

Response

Although capacity building in L&M and all other assessments involved 20 facilities, the 2018 BRN star rating assessment was not done in one HC. We have added a sentence to clarify it in the results – section. 

15. In Figure 3, what does the dotted line represent? Is it the overall trend in the referral rate in the intervention HCs?

Response

The dotted line represented the linear trend in the referral rates in the intervention HCs. However, to avoid confusion, we have removed it.

16. Results, Utilization of CEmONC services, sentences: “For instance, the mean monthly deliveries at Gairo and St. Joseph HCs increased from 71 (67 – 76) to 137 (124 - 150) during intervention period, and from 48 (41 – 55) to 129 (116 - 143) respectively. The mean monthly deliveries at Kibati and Ngerengere increased from 21 (18 – 23) to 34 (30 – 37) during intervention period, and from 26 (23 - 28) to 33 (31 – 36) respectively.” Specify the meaning of the numbers in bracket.

Response

The numbers in the brackets signify 95% CI. These have been added in the manuscript.

17. Results, Quality of CEmONC services: What were the primary causes of the maternal deaths and the avoidable/modifiable factors associated with them (at least in the intervention HCs).

Response

We have added the following sentences in the manuscript. “During these periods (baseline and during the intervention) 40 maternal deaths occurred. The primary causes of these deaths during the intervention period were postpartum haemorrhage 8, pre/ eclampsia 5, puerperal sepsis 3, complications of anaesthesia 2, uterine rupture 2, severe anaemia in pregnancy 2 and antepartum haemorrhage 1. The causes of deaths were not established in 13 maternal deaths that occurred at baseline, and four that occurred during the intervention in the control HCs because of inadequate records keeping. Avoidable factors were determined in only 87% i.e., 20 cases out of 23 deaths because the case files for the other 3 had inadequate information. Delay to provide appropriate care after reaching the facilities was identified in 80% i.e., 16 of 20 facilities. A delay in seeking treatment/ reaching the facility was found in 55% i.e., 11 out of 20 deaths.”

18. Results, The requirements and costs of scaling up of CEmONC services in health centres, sentence: “Detailed findings on the requirements and costs for scaling up CEmONC in health centres in Tanzania are reported elsewhere.” Reference 19 was cited by the authors. However, the word Tanzania could not be found in reference 19 (i.e. Model of Care Initiative in Nova Scotia (MOCINS): Final Evaluation Report).

Response

We have replaced the reference with the appropriate one.

19. Discussion, Health systems strengthening for maternal and newborn health care, sentence: “Strengthening leadership and management at the health facility district and regional health system levels resulted in strengthened health systems building blocks, which are vital for provision of effective services.” It is preferrable to use the words “was associated with” rather than “resulted in.”

Response

Replacement done. Thank you

20. Discussion, Scale up of CEmONC services, first paragraph, sentence: Improving the availability and access to comprehensive emergency obstetric and neonatal care services resulted in a marked increase in utilization of services (including women with obstetric complications) and reduced referral rates to distant hospitals in intervention centres. It is preferrable to use the words “was associated with” rather than “resulted in.” This is because the improvements could have been due to other factors such as changes in human migration and population. These may explain some of the outcomes in the control HCs.

Response

Replacement done. Thank you

21. Discussion, Scale up of CEmONC services, second paragraph, sentence: “Maternal mortality also dropped in the two control centres but they continued to refer pregnancies at the same rate to secondary hospitals, suggesting little change in the complexity of pregnancies and deliveries they were managing.” The data on complexity of the cases managed at the intervention and control HCs were not presented in the result section.

Response

Thank you. By saying the complexity of pregnancies and deliveries the authors meant complicated pregnancies managed at the health facilities. In the results in table 3, the authors presented a summary of women who had obstetric complications (maternal morbidities) at baseline and during the intervention period. In the Results, Utilization of CEmONC services (sub-section): we also presented figures of women with obstetric complications referred to the nearby district hospital. We thought that the data on maternal morbidities and referred pregnant women provided a snapshot of complexity of pregnancies and that somehow justify our statement. 

22. Discussion, Scale up of CEmONC services, fourth paragraph, sentence: “Findings from previous and current studies coupled with effective knowledge translation strategies, engagement, political will and commitment resulted in a nation-wide scale up of CEmONC services in public health centres.” Delete “and current.”

Response

The words are deleted

23. Discussion, “Addressing the fear of the unknown: the cost of scaling up CEmONC services,” sentence: “The requirements and related costs reported in this study fill the existing vacuum of science and knowledge.” Delete this sentence.

This is because the authors also wrote in the results section that “Detailed findings on the requirements and costs for scaling up CEmONC in health centres in Tanzania are reported elsewhere.” Therefore the index report/study can’t be filling any vacuum in knowledge. Additionally, there are previous studies on the cost of scaling up a health facility in low- and middle-income countries.

Response

The sentence is deleted

24. Discussion, Limitations of the study: Acknowledge that other factors such as changes in human migration and population could have affected the findings in both intervention and control HCs.

Response

Thank you. These factors have been added in the list of limitations

25. Conclusion: “Integration of leadership and managerial capacity building, with CEmONC-specific interventions has resulted in health systems strengthening and improved quality of services.” It is preferrable to use the words “was associated with” rather than “resulted in.”

Response

The words are replaced. Thank you

---

## [Editor Report · Decision Letter 3]

28 Jun 2022

Scale up and strengthening of comprehensive emergency obstetric and newborn care in Tanzania

PONE-D-21-37381R3

Dear Dr. Nyamtema,

We’re pleased to inform you that your manuscript has been judged scientifically suitable for publication and will be formally accepted for publication once it meets all outstanding technical requirements.

Kind regards,

Nnabuike Chibuoke Ngene, Dip HIV Med; MMed(FamMed); FCOG; MMed(O&G); Ph.D

Academic Editor

PLOS ONE
---

## [Editor Report · Acceptance letter]

1 Jul 2022

PONE-D-21-37381R3 

Scale up and strengthening of comprehensive emergency obstetric and newborn care in Tanzania 

Dear Dr. Nyamtema:

I'm pleased to inform you that your manuscript has been deemed suitable for publication in PLOS ONE. Congratulations! Your manuscript is now with our production department. 

Kind regards, 

on behalf of

Dr. Nnabuike Chibuoke Ngene 

Academic Editor

PLOS ONE